# Antibiofilm Effect of Biogenic Silver Nanoparticles Combined with Oregano Derivatives against Carbapenem-Resistant *Klebsiella pneumoniae*

**DOI:** 10.3390/antibiotics12040756

**Published:** 2023-04-14

**Authors:** Sara Scandorieiro, Franciele Maira M. B. Teixeira, Mara C. L. Nogueira, Luciano A. Panagio, Admilton G. de Oliveira, Nelson Durán, Gerson Nakazato, Renata K. T. Kobayashi

**Affiliations:** 1Laboratory of Basic and Applied Bacteriology, Department of Microbiology, Center of Biological Sciences, Universidade Estadual de Londrina, Londrina 86057-970, Brazil; sarascandorieiromicro@gmail.com (S.S.); gnakazato@uel.br (G.N.); 2Laboratory of Innovation and Cosmeceutical Technology, Department of Pharmaceutical Sciences, Center of Health Sciences, Hospital Universitário de Londrina, Londrina 86038-350, Brazil; 3Department of Dermatological, Infectious and Parasitic Diseases, Faculdade de Medicina de Sao Jose do Rio Preto, São José do Rio Preto 15090-000, Brazil; mairabiomed@hotmail.com (F.M.M.B.T.); ml.nogueira@famerp.br (M.C.L.N.); 4Laboratory of Medical Mycology and Oral Microbiology, Department of Microbiology, Center of Biological Sciences, Universidade Estadual de Londrina, Londrina 86057-970, Brazil; lapanagio@uel.br; 5Laboratory of Microbial Biotechnology, Department of Microbiology, Center of Biological Sciences, Universidade Estadual de Londrina, Londrina 86057-970, Brazil; admilton@uel.br; 6Laboratory of Electron Microscopy and Microanalysis, Center of Biological Sciences, Universidade Estadual de Londrina, Londrina 86057-970, Brazil; 7Institute of Biology, Universidade Estadual de Campinas, Campinas 13083-862, Brazil; nelsonduran1942@gmail.com

**Keywords:** green nanotechnology, oregano oil, carvacrol, thymol, anti-quorum sensing, crystal violet, MTT, scanning electron microscopy

## Abstract

Resistant bacteria may kill more people than COVID-19, so the development of new antibacterials is essential, especially against microbial biofilms that are reservoirs of resistant cells. Silver nanoparticles (bioAgNP), biogenically synthesized using *Fusarium oxysporum*, combined with oregano derivatives, present a strategic antibacterial mechanism and prevent the emergence of resistance against planktonic microorganisms. Antibiofilm activity of four binary combinations was tested against enteroaggregative *Escherichia coli* (EAEC) and *Klebsiella pneumoniae* carbapenemase-producing *K. pneumoniae* (KPC): oregano essential oil (OEO) plus bioAgNP, carvacrol (Car) plus bioAgNP, thymol (Thy) plus bioAgNP, and Car plus Thy. The antibiofilm effect was accessed using crystal violet, MTT, scanning electron microscopy, and *Chromobacterium violaceum* anti-quorum-sensing assays. All binary combinations acted against preformed biofilm and prevented its formation; they showed improved antibiofilm activity compared to antimicrobials individually by reducing sessile minimal inhibitory concentration up to 87.5% or further decreasing biofilm metabolic activity and total biomass. Thy plus bioAgNP extensively inhibited the growth of biofilm in polystyrene and glass surfaces, disrupted three-dimensional biofilm structure, and quorum-sensing inhibition may be involved in its antibiofilm activity. For the first time, it is shown that bioAgNP combined with oregano has antibiofilm effect against bacteria for which antimicrobials are urgently needed, such as KPC.

## 1. Introduction

This manuscript reports, for the first time, the antibiofilm effect of eco-friendly binary combinations composed of oregano compounds (OEO, Car, and Thy), which is considered GRAS (Generally Recognized as Safe) [1], and biogenic silver nanoparticles (bioAgNP) that is produced by green nanotechnology with a low-cost method using *F. oxysporum* components [2]. The present study shows that they eradicate preformed biofilm and prevent its formation by enteroaggregative *E. coli* (EAEC 042) and carbapenemase (KPC)-producing *K. pneumoniae*. Biofilm-associated pathogenic bacteria are a serious problem in several fields [3].

Biofilm formation on food and its packaging cause equipment impairment, food spoilage, and diseases [4,5]. There are more than 200 foodborne illnesses, making 1 in 10 people sick worldwide and causing 420,000 deaths each year; in addition, it has negative consequences for socioeconomic development by overloading health systems and damaging the national economy, tourism, and trade [5,6]. The food industry incorporates additive (such as nisin, natamycin, and sorbates) in their products to reduce contamination and to act as preservatives; however, previous studies reported the potential genotoxic effect of synthetical antimicrobials such as nisin and natamycin. Therefore, oregano derivatives are alternative additives to replace conventional ones, as Campos et al. (2022) have shown [7]. Studies also show that OEO [8] or bioAgNP [9] can be incorporated in packaging to increase the shelf life of food products and to avoid their contamination with pathogens.

Bacterial biofilms have a huge negative impact on human health, causing infections in the sinuses and middle ear, dental plaque, periodontitis, endocarditis, osteomyelitis, wound infections, and infections related to devices such as pacemakers, several types of catheters, joint prostheses, or implants (e.g., heart valves, dentures, and contact lenses) [10,11]. Biofilm-forming bacteria also have large negative impacts on the veterinary field, causing huge economic losses (diseases in poultry and ruminants), besides making pets sick [12]. Multidrug-resistant bacteria are the greatest challenge for the treatment of human and other animal infections since such microorganisms are resistant to the majority of available antimicrobials; some bacteria are called pan-resistant because they show resistance to all available antibiotics, and biofilm strongly contributes to this resistance profile [13].

Conventional antimicrobials have difficulty reaching the biofilm cells because their diffusion is limited by biofilm structure. The exopolysaccharide matrix is a physical barrier which makes it difficult for antibiotics to penetrate the deep biofilm layers; additionally, drug molecules also may interact with matrix components, resulting in a slow penetration rate and, consequently, microbial resistance profile [3,13,14]. Biofilm has a heterogeneous nature, with metabolic diversity that may influence antimicrobial activity. There is a nutrient and oxygen gradient in the biofilm environment, which results in active cells in superficial biofilm layers and anaerobic, less active cells in the bottom [15,16]. Many antibiotics only affect bacteria during their active metabolic stage (e.g., betalactams, aminoglycosides, tetracyclines, chloramphenicol, lincomycin, clindamycin, macrolides, quinolones, rifampin, among others), presenting an effect on synthetic processes of proteins, cell wall, and nucleic acid [15,17]. Efflux pumps are important for biofilm formation since the secretion of EPS and quorum-sensing molecules is extremely needed in this microbial lifestyle. However, these systems also can extrude harmful molecules as antibiotics from within the cell and collaborate for the resistance profile of biofilms [16,18].

Besides biofilm-intrinsic resistance, other mechanisms of resistance (antibiotic target modification, drug-inactivating enzyme, efflux pumps, alteration of porin expression, and others) may allow pathogenic bacteria to be tolerant to all classes of conventionally available antimicrobials [13,16]. The high cellular density and cell proximity in biofilms increase the chances of genetic exchanges, including between different species, making biofilms antibiotic resistance reservoirs [13,19].

Currently, more than 1.2 million people die in a year due to multidrug-resistant infections [20]. If no action is taken, it is estimated that this mortality rate will increase alarmingly by 2050, causing one death every three seconds and 10 million deaths a year [21]. The death rate is higher than COVID-19, which killed more than 6.8 million people in a period of three years (data referring to February 2023) [22]. The Centers for Disease Control and Prevention [23] and World Health Organization [24] have highlighted the urgent need for research and development of alternative antimicrobials with the potential to combat resistant bacteria and also prevent the emergence of resistance. In addition, respiratory viral diseases, such as severe acute respiratory syndrome (SARS) and COVID-19, predispose the patient to secondary bacterial infections [25]. Therefore, the current pandemic scenario reinforces the need to develop new antibiotics to treat bacterial infections.

Among the alternatives, essential oils and their components stand out. Studies have shown the antibiofilm activity of oregano essential oil (OEO) and its main bioactive, carvacrol (Car) and thymol (Thy), to combat Gram-positive and Gram-negative pathogens, including multidrug-resistant strains [26,27,28,29,30,31], showing potential to be applied as antimicrobials in clinical and hospital sectors. Substances of plant origin have successfully been used in traditional medicine since ancient times, such as tea or herbal infusions, for example. Currently, around 300 essential oils are available in the market; they are natural compounds already incorporated in several products for human consumption because they are safe at small dosages and under specific recommendations [32], so the study and application of oregano essential oils as an antibiofilm agent is quite viable. Despite their excellent antimicrobial activity, oregano terpenoids have a strong taste and smell that can be attenuated by combining them with other compounds [33].

Ancient civilizations such as the Persians, Phoenicians, Greeks, Romans, and Egyptians have made use of silver antimicrobial properties to preserve food and water and treat eye infections and wounds [34,35]. Nanotechnology has allowed reviving the use of this metal as an antimicrobial since silver nanoparticles (AgNP) have been used for over 100 years [36] and are currently incorporated into various products and processes of our routine (such as formulation of surface cleaners, water disinfection processes, wound dressing, several materials for cutaneous infections, coating for bone implants, and catheters) [37,38,39]. The antibiofilm activity of AgNP has been shown by several studies [40,41,42] since silver has high oxidation-reduction potential, and the large surface area of its nanoparticle form provides them broad antimicrobial properties [36,43]. Resistance to ionic silver and bioAgNP is already reported [44,45,46], and the antimicrobial combination is a strategy already practiced in clinical settings to prevent the emergence of resistant strains.

Our research group has shown that combinations containing OEO, Car, Thy, and bioAgNP present a strategic mechanism of action and prevent the emergence of resistance in planktonic bacteria, exhibiting synergistic or additive and fast activity at low concentrations against Gram-positive and Gram-negative bacteria, including multidrug-resistant strains such as carbapenem-resistant bacilli (*Escherichia coli*, *Klebsiella pneumoniae*, *Pseudomonas aeruginosa*, and *Acinetobacter baumannii*) and methicillin-resistant *Staphylococcus aureus* (MRSA) [31,47]. It is important that antimicrobial agents act against bacteria both in the free and aggregated state because the biofilm is a reservoir of planktonic cells that are released from their sessile stage and contribute to the persistence of the infection, despite their biofilms being the most prevalent life form in nature and being more resistant to antimicrobials [48]. Thus, the present study shows the antibiofilm effect of oregano derivatives and bioAgNP in combination, indicating that such combinations act widely to combat bacterial resistance.

## 2. Results

### 2.1. Biogenic Silver Nanoparticles (bioAgNP) Characterization Biogenic Silver Nanoparticle

Fungal-free solution without AgNO_3_ was pale yellow. After adding silver salt, the solution color changed to brownish, and its color intensity increased over the course of time while incubated at 28 °C, suggesting bioAgNP formation. The nanoparticle plasmonic band also indicated that bioAgNP were synthetized; the bioAgNP exhibited a surface plasmon-resonance peak centered at 410 nm, while pale yellow fungal-free solution (negative control) showed no absorption peak at this wavelength; the UV-visible absorbance spectra are shown in Figure 1A.

Photon correlation spectroscopy and scanning microscopy analysis confirmed nanoparticle formation. The scanning electron microscopy (SEM) micrograph of bioAgNP shows the spherical shape of nanoparticles (Figure 1B). The average bioAgNP diameter and zeta potential were 84.10 nm and −15.9 mV, respectively. Nanoparticle size and zeta-potential distributions are shown in Appendix A, respectively. Energy-dispersive X-ray fluorescence spectrometer analysis confirmed the presence of Ag in nanoparticle samples (silver concentration obtained by linear regression is shown in Appendix A).

### 2.2. Sessile Minimal Inhibitory Concentration (SMIC) of Compounds Alone against Both Initial Stage of Biofilm Formation and Preformed Biofilm

As shown in Appendix A, OEO, Car, Thy, and bioAgNP alone significantly prevented biofilm formation (Appendix A) and decreased metabolic activity of preformed biofilm (Appendix A) of both EAEC 042 and KPC-producing *K. pneumoniae*, compared to untreated control (*p* < 0.05, Kruskal-Wallis test). SMIC values were determined at 95% or greater inhibition (SMIC_≥95_) of metabolic activity compared to untreated positive control. Table 1 indicates SMIC_≥95_ of each antimicrobial alone for EAEC O42 and KPC-producing *K. pneumoniae* at both the initial stages of biofilm formation and preformed biofilm; SMIC_≥95_ values were determined with the MTT method.

OEO, Car, Thy, and bioAgNP alone significantly prevented biofilm formation and decreased metabolic activity of preformed biofilm of both EAEC 042 and KPC-producing *K. pneumoniae*, compared to the untreated control (*p* < 0.05, Kruskal–Wallis test). As shown in Appendix A (data for biofilm at the initial stage of formation and preformed biofilm, respectively), all four antibacterials reduced at least 95% of biofilm viability and caused at least 90% reduction in total biofilm biomass.

### 2.3. Sessile Minimal Inhibitory Concentration (SMIC) of Compounds in Combinations against Both the Initial Stage of Biofilm Formation and Preformed Biofilm

For preformed biofilm or its initial stage of formation, the four combinations showed improved antibiofilm activity against EAEC and KPC strain by reducing SMIC_≥95_ of each compound in association (Table 2).

We highlight the combination of Thy plus bioAgNP that resulted in SMIC_≥95_ reductions of 50% for Thy and 75% for bioAgNP compared to compounds alone against EAEC biofilm formation. For its preformed biofilm, this combination reduced SMIC_≥95_ by 75% for Thy and by 87.5% for bioAgNP.

For KPC-preformed biofilm, bioAgNP at 3.94 µg/mL and Thy at 0.12 mg/mL (both values represent 0.5 × SMIC_≥95_ individually) have not been tested in combinations since both concentrations alone already show a great antibiofilm effect, reducing metabolic activity by around 80%; Thy plus bioAgNP reduced SMIC_≥95_ of both antimicrobials by 50% against biofilm on the initial stage of formation.

In a few cases, as shown in Table 2, combinations did not cause SMIC_≥95_ reductions, but they still showed a better effect than antimicrobial alone by causing a greater decrease in biofilm metabolic activity and/or total biomass compared to individual treatments (Section 2.4). For example, Car associated with bioAgNP did not reduce SMIC_≥95_ of compounds against the KPC strain; however, this combination showed an improved effect compared to separated compounds.

### 2.4. Antibiofilm Effect of Binary Combinations Compared to Antimicrobials Individually against Preformed Biofilm in Microtiter Plates and Its Initial Stage of Formation (Reduction in Biomass and Metabolic Activity)

All tested combinations (OEO plus bioAgNP, Car plus bioAgNP, Thy plus bioAgNP, and Car plus Thy) prevented biofilm formation and inhibited the growth of preformed biofilm by both EAEC 042 and carbapenem-resistant K. pneumoniae. For each bacterium, comparative analysis among each combination and untreated control showed that double antimicrobial treatments significantly reduced total biofilm biomass and biofilm metabolic activity (*p* < 0.05, Kruskal–Wallis test), as shown in Appendix A for the initial stage of biofilm formation and in Appendix A for preformed biofilm.

All four combinations showed improved antibiofilm activity compared to antimicrobials individually against preformed biofilm and its initial stage of formation. Each combination and its antimicrobials alone at the same concentrations showed that double antimicrobial treatments significantly caused a greater reduction in total biofilm biomass and biofilm metabolic activity in percentage (*p* < 0.05, Kruskal–Wallis test), as shown in Table 3, Table 4, Table 5 and Table 6. Both Table 3 and Table 4 present the combinatory effect of antimicrobials on biofilm at the initial stage, and Table 5 and Table 6 present the combination antibiofilm effect against preformed biofilm.

To allow comparison between combined and individual treatments, only concentrations that alone did not have or had a small antibiofilm effect were chosen. We highlight that Thy and bioAgNP, both alone at a subinhibitory concentration (lower than SMIC_≥95_), did not inhibit EAEC and KPC biofilm formation, respectively, but both antimicrobials in binary combinations with other compounds prevented biofilm formation.

### 2.5. Scanning Electron Microscopy Study of Preformed Biofilm Treated with Compounds Alone and in Combination

Figure 2A–I, Figure 3A–I and Figure 4A–I show the effect of antibacterials individually (OEO, Car, Thy, and bioAgNP) and in combination (OEO plus bioAgNP, Car plus bioAgNP, Thy plus bioAgNP, and Car plus Thy) on preformed biofilm of EAEC 042 in the glass surface.

Figure 2A–I show the biofilm amount on the glass surface since all images were taken at magnification 1600× and allow surface-wide analysis. The untreated sample (Figure 2A) shows a slightly high-density biofilm, with a great number of cells, bacterial aggregation, formation of microcolonies, and architecture at the initial stage of organization. Figure 2A represents a healthy biofilm at an early stage of maturation since it does not have high biomass density, and its three-dimensional architecture is still under development (Figure 2A). Four treated samples show damaged biofilm, with less dense biomass, no early architecture organization, extensively decreases in cell density, and smaller and scattered sparsely cellular aggregates, compared to the untreated control; Thy-treated (Figure 2D), bioAgNP-treated (Figure 2E), Car plus bioAgNP-treated (Figure 2G), and Thy plus bioAgNP-treated (Figure 2H). The OEO-treated sample (Figure 2B) and OEO plus bioAgNP-treated sample (Figure 2F) show less dense biofilm, with slightly less cellular density and bacterial aggregation compared to the untreated sample. Car-treated sample (Figure 2C) and Car plus Thy-treated sample (Figure 2I) show biofilm with a high cellular density similar to untreated control.

Figure 3A–I show the exopolysaccharide matrix and intercellular connections since images were taken at magnification 10,000×. The untreated sample (Figure 3A) shows the initial organized three-dimensional architecture, in which extracellular fibril is distributed in an orderly manner, presenting intercellular networks that is typical of healthy biofilm. All antimicrobials affected the development of biofilm architecture, which seems unstructured, resulting in a less dense matrix and loss of cell connections compared to untreated control. Such damages are more intense in the following treated samples: Thy (Figure 3D), bioAgNP (Figure 3E), Car plus bioAgNP (Figure 3G), and Thy plus bioAgNP (Figure 3H). OEO-treated (Figure 3B), Car-treated (Figure 3C), and Car plus Thy (Figure 3I) show more cellular density compared to control and other treatments; however, intercellular networks are extremely poor.

Figure 4A–I consist of images at high magnification (20,000×) that show details of morphological changes and deformations in EAEC 042-treated cells and no alterations in untreated cells. The untreated sample (Figure 4A) shows EAEC cells with an intact surface, typical size, and rod shape. OEO-treated (Figure 4B) and Thy-treated (Figure 4D) samples show cells with morphological alterations, such as the sinking of the bacterial cell wall and irregular wrinkled surface. BioAgNP-treated biofilm (Figure 4E) shows cells with reduced size compared to untreated control and deformed cells with an irregular wrinkled surface. The OEO plus bioAgNP sample (Figure 4F) shows altered cells with an irregular wrinkled surface. Car plus bioAgNP (Figure 4G) and Thy plus bioAgNP (Figure 4H) samples show cells with reduced size and irregular wrinkled surfaces, compared to untreated control. The Car-treated sample (Figure 4C) and Car plus Thy sample (Figure 4I) show cellular population without remarkable morphological alterations.

### 2.6. Scanning Electron Microscopy Study of 24 h-Biofilms of EAEC and KPC Strains

Figure 5 consists of scanning electron micrographs (1000×) of EAEC 042 and KPC-producing *K. pneumoniae* biofilms grown for 24 h in glass slides. EAEC (Figure 5A) exhibits biofilm in an immature phase but is more developed than *K. pneumoniae* (Figure 5B). After 24 h of growth, *E. coli* biofilm already shows very initial development of three-dimensional architecture, with cellular aggregates and a remarkable amount of EPS. With the same growth time, *K. pneumoniae* shows non-aggregated cells and an extremely minimal amount of EPS, still without the three-dimensional structure characteristic of biofilm.

### 2.7. Effect of Compounds on Violacein Production

The mean MIC values of antibacterials against *Chromobacterium violaceum* are as follows: 0.15 mg/mL for OEO, 0.08 m/mL for Car, 0.12 mg/mL for Thy, and 23.13 µM for bioAgNP. MIC values were determined by the Clinical and Laboratory Standards Institute standard methodology (with initial bacterial inoculum at 5 × 10^5^ CFU/mL) to determine the subinhibitory concentrations that would be tested in the violacein assay. The chosen subinhibitory concentrations of antimicrobials are indicated in Appendix A. Violacein reduction is expressed in percentage and compared to untreated control. Non-treated control is defined as 100% of violacein production.

Oregano-derivative antibacterials and bioAgNP, individually and in combination, reduced violacein production and did not inhibit *C. violaceum* growth. All treated *C. violaceum* samples and untreated samples are similar with regard to the number of viable cells (approximately 10^9^ CFU/mL). However, violacein production was reduced by 93% (OEO), 94% (Car), 92% (Thy), 81% (bioAgNP), and 95% (Thy plus bioAgNP) compared to non-treated *C. violaceum* (Appendix A). Figure 6A qualitatively shows that OEO, Car, Thy, bioAgNP, and Thy plus bioAgNP have inhibitory effects on violacein production by *C. violaceum* since all treated bacterial cultures visually lack violet pigment or present dramatically reduced violet color compared to untreated control. Figure 6B quantitatively shows significant differences among OEO, Car, Thy, bioAgNP, Thy plus bioAgNP, and untreated control in terms of violacein amount (*p* < 0.05, Kruskal–Wallis test).

## 3. Discussion

This study shows the antibiofilm effect of oregano-derived compounds and bioAgNP (biologically synthesized using *F. oxysporum* components) against Enterobacteriaceae strains, such as EAEC 042 and carbapenem-resistant *K. pneumoniae*. Oregano derivatives combined with bioAgNP present action against the KPC strain, which is identified in the WHO priority pathogen list for which effective antimicrobials are urgently needed [24]. Additionally, such combinations may prevent the emergence of resistance and minimize undesirable organoleptic effects of oregano terpenoids since association may require a lower concentration of each compound compared to their use alone [31]. The four binary combinations presented in this study are eco-friendly since oregano compounds are considered GRAS (Generally Recognized as Safe) [1], and bioAgNP root synthesis is less toxic than chemically synthesized nanoparticles because chemical reagents are not used as reducing or stabilizing agents [49]. Furthermore, *F. oxysporum*–bioAgNP is stable for several months due to protein capping, which occurs in the biogenic process, as seen by electron microscopy [50].

We report the antibiofilm action of these compounds against both the initial stage of formation and preformed biofilm. Biofilms are heterogeneous in their structure, organization, and metabolic characteristics; their life cycle complexity must be considered for evaluating the results of antibiofilm assays since antimicrobials can exhibit effects against biofilm at different stages [51,52,53].

In this study, three methodologies were used (crystal violet staining, MTT assay, and SEM) for accessing biofilm total biomass, its viability, and structure, respectively; in addition, an initial study about the anti-quorum-sensing effect of compounds was carried out using *C. violaceum*. The present data highlight the importance of using combined methodologies to access the antibiofilm activity of compounds; each method has its advantages and limitations and evaluates a specific aspect of biofilm, and in combination, they allow more reliable conclusions [51,54].

The violet crystal technique was crucial in our analysis of biofilm at an early stage of formation since the MTT assay alone would not show the efficiency of some antimicrobials to prevent biofilm formation. Some treatments prevent biofilm formation (as they reduce total biomass production measured by the violet crystal) but possibly cause stress in bacterial populations because the MTT test shows treated samples with high metabolic activity similar to the non-treated control.

However, MTT assay was essential for our preformed biofilm study since this technique was more sensitive than violet crystal to show antimicrobial activity against biofilm at the advanced stage of development. The violet crystal test was not feasible for detecting preformed biofilm biomass, but the microscopic technique allowed this analysis to be successful (Figure 2, Figure 3 and Figure 4).

This research also showed the effect of compounds on the growth of biofilm in a polystyrene microtiter plate (Table 3, Table 4, Table 5 and Table 6) and glass slide (Figure 2, Figure 3 and Figure 4). Different surfaces and environments influence biofilm growth and also may impact biofilm susceptibility to antibiotics [53,55,56].

OEO, Car, Thy, and bioAgNP individually prevented EAEC 042 biofilm formation. All four compounds alone at SMIC_≥95_ also inhibited planktonic cell growth of *E. coli*. Only OEO, Car, and bioAgNP inhibited EAEC biofilm formation at subinhibitory concentrations for planktonic cells (bellow SMIC_≥95_) (Table 3), and it suggests that these three compounds show the effect on biofilm formation by interfering both in planktonic cells growth and also in specific pathways of sessile cells. In contrast, Thy-antibiofilm activity may rely on action against planktonic cells since it did not cause a reduction in biofilm biomass and its viability at a concentration below SMIC_≥95_. Planktonic cells and biofilm life styles of single species express different genes, consequently accomplishing different phenotypic profiles [57,58]. Biofilm inhibition at subinhibitory concentrations (for planktonic cells) might be due to the inhibitory effect on the expression of genes related to motility and biofilm formation or the effect on specific biofilm structures and metabolic paths [16,59,60]. In agreement with our data, other studies also reported that OEO, Car [28,61,62], and bioAgNP [40,63,64] prevent *E. coli* biofilm formation.

For KPC-producing *K. pneumoniae*, all four tested antibacterials individually prevented biofilm formation; SMIC_≥95_ values also inhibited planktonic cell growth. At subinhibitory concentrations for planktonic cells, only OEO, Car, and Thy inhibited the biofilm formation of carbapenem-resistant *K. pneumoniae* (Table 4). It suggests that oregano compounds show an effect on biofilm formation by interfering both in planktonic cell growth and also in specific pathways of biofilm lifestyle since they reduced total biofilm biomass and its metabolic activity compared to the untreated bacterium. In comparison, bioAgNP–antibiofilm activity may rely on action against planktonic cells since these nanoparticles did not cause a reduction in biofilm biomass and its viability at a concentration below SMIC_≥95_. Some researchers also showed that OEO, Car, and Thy inhibited *K. pneumoniae* biofilm formation [26,30] in agreement with our data.

In this study, EAEC 042 was more sensitive to bioAgNP than KPC-producing *K. pneumoniae* since these metal nanoparticles prevented biofilm formation at SMIC_≥95_ of 0.49 µg/mL for *E. coli* and 1.97 µg/mL for *K. pneumoniae*. The literature data show that other biogenic silver nanoparticles prevent biofilm formation by several bacterial species, including *E. coli*, *K. pneumoniae,* and *Pseudomonas aeruginosa*, with a wide range of SMIC [40,63,65,66]. Unlike our data, some studies show that silver nanoparticles, even at subinhibitory concentrations, inhibited biofilm formation by *K. pneumoniae* [41,63,67]. Thus, comparison of results is difficult since the effective concentration of bioAgNP varies in each study because of nanoparticle diversity in terms of size, morphology, composition, stabilizing agents, and surface charge; furthermore, the use of different techniques for nanoparticle characterization and microbiological analysis may influence conclusion regarding the antimicrobial activity [68,69,70,71]. To reduce factors that limit the comparison of results between different studies, we highlight the importance of standardization of bioAgNP characterization and their microbiological assays, specifically with regard to antibiofilm assays [52,69]. Moreover, different bacterial strains used in several studies may have structural and metabolic differences that make them more or less sensitive to such compounds [28,40,58,72].

At concentrations lower than SMIC_≥95_, OEO and Car prevented biofilm formation of EAEC and *K. pneumoniae*, but both compounds seem to act in different ways against both strains. For KPC-producing *K. pneumoniae*, the two oregano compounds reduced both biofilm biomass and biofilm viability. In the case of EAEC, both OEO and Car reduced total biomass, but it seems they caused bacterial stress response since the treated biofilm showed high metabolic activity similar to the non-treated control. It is known that during acid stress, *E. coli* upregulates some components of the electron transport chain (e.g., several dehydrogenases); under normal growth, such enzymes are involved in generating proton motive force by redox reactions with exportation of protons from the cells [73]. The literature indicates that essential oil can acidify the bacterial cytoplasm, which would justify the increase in metabolic activity detected in this study [74,75]. Furthermore, the mechanism of biofilm formation varies between *E. coli* and *K. pneumoniae* [58,72,76]; these differences may contribute to both bacteria responding differently to different treatments. *K. pneumoniae* initial colonization is a more passive process compared to *E. coli*; probably, it happens due to lack of motility in *K. pneumoniae* whose cells are less metabolically active at stages of attaching to surfaces and become progressively active in mature biofilm [77]. Such metabolic differences at the initial stages of biofilm formation may explain why OEO or Car treatments increased EAEC cell viability and decreased it in KPC-producing *K. pneumoniae*. García-Heredia et al. [28] reported that OEO and Car inhibited biofilm formation by EAEC 042, but both compounds did not prevent EAEC O104:H4 biofilm formation. It indicates that EAEC strains show a difference in their genomic regulation, suggesting that responses to oregano derivatives are not only compound-dependent but may also depend on strain-to-strain variations, in agreement with our results.

The present results suggest that OEO, Car, Thy, and bioAgNP may show specific inhibiting effects on different bacterial species. Several mechanisms may drive their antibiofilm properties, such as reducing fimbriae production, decreasing swarming motility, reducing flagellar biosynthesis, quorum-sensing interruption, inhibition of efflux pumps, and others. This study showed that all oregano compounds and bioAgNP reduced violacein production by *C. violaceum* (Figure 6) in agreement with the literature [32,33,78,79,80,81,82,83], suggesting that disruption of quorum sensing is one of the ways by which they prevent biofilm formation since the production of purple pigment violacein is directly linked to quorum sensing [32]. However, the antibiofilm mechanisms of these compounds must be investigated in detail to evaluate how such compounds modulate the expression of genes that are involved in biofilm formation, for example.

Mature biofilms can protect bacteria living inside against several adverse environmental influences and conditions. Antibiotics or disinfectants frequently fail to remove biofilms from biological or non-biological surfaces, which can represent a source of recurrent infections [84]. Biofilm bacteria often tolerate antibiotics at concentrations 10–10,000-fold greater than planktonic cells [32,85]. For eradicating successfully mature biofilms, it is necessary that antimicrobials penetrate into the aqueous channels of biofilms [84]; OEO and their main components (Car and Thy), despite being lipophilic-volatile substances, caused at least a 95% reduction in metabolic activity of preformed biofilm by both EAEC and KPC-producing *K. pneumoniae* in a microtiter plate (Table 1).

Preformed biofilms and biofilm under formation condition of both bacterial strains showed similar susceptibility to oregano-derived compounds; for EAEC, SMIC_≥95_ values were two-fold greater against pre-established biofilm than biofilm at an early stage of formation. For carbapenem-resistant *K. pneumoniae*, SMIC_≥95_ values were the same for biofilm at both stages. Our previous study showed that OEO, Car, and Thy present similar minimum inhibitory concentration (MIC) against bacterial planktonic cells, including multidrug-resistant strains [31]. In agreement with present data, Reichling [84] and Yadav et al. [86] highlight that several essential oils and individual oil compounds show similar MIC values for planktonic cells and their biofilm. Unlike our data, some studies reported that planktonic bacterial cells are more sensitive to OEO, Car, and Thy than their biofilms [84,87,88]; some researchers have found that Thy inhibited biofilm formation by *E. coli* [29,61,62]. Result variances among different studies may occur because oregano compounds derive from plants and undergo variations in their chemical composition, which are dependent on climatic and geographical factors, and also extraction methods [32,89].

The bioAgNP also eliminated at least 95% of preformed biofilm by both tested strains. However, preformed biofilms were less susceptible to bioAgNP than biofilm at an early stage of development (SMIC_≥95_ values are shown in Table 1), in agreement with other studies which suggested that biofilm greater resistance might be partially attributed to nanoparticle aggregation and retarded silver ion and particle diffusion [90,91,92]. For EAEC 042, bioAgNP SMIC_≥95_ was 32-fold higher against preformed biofilm than biofilm under formation conditions. For carbapenemase-producing *K. pneumoniae*, bioAgNP SMIC_≥95_ was four-fold greater against pre-established biofilm than biofilm at the initial stage formation.

This present study also showed that the preformed biofilm of *E. coli* is more tolerant to bioAgNP than *K. pneumoniae* pre-established biofilm since bioAgNP SMIC_≥95_ is higher for EAEC (Table 1); the structural difference of biofilm between the two species may explain this difference in susceptibility to bioAgNP. SEM micrographs of untreated biofilms (24 h of formation) of both bacterial species (Figure 5) showed that *K. pneumoniae* presented a more youthful biofilm with lower cell density, little secreted EPS, and less cell aggregation compared to EAEC. Glycocalyx and the EPS matrix of biofilms act as biding sites and limit antimicrobial diffusion through the matrix, reducing drug access to sessile cells [3,93]. In addition, bioAgNP treatment may decrease EPS production in *K. pneumoniae* [41], contributing to the greater sensitivity of this strain to bioAgNP.

However, oregano terpenoids and silver nanometal exhibit features that may limit their applications as antimicrobials. OEO, Car, and Thy present high volatility and strong organoleptic features [33,94], and bacteria easily develop resistance to silver nanoparticles [44,45,46,95]. We showed previously that *E. coli* develops resistance to bioAgNP after 12 days of daily treatment [31]. Thus, in order to solve these problems and expand the possibilities for these compound applications, our research group proposes the association between oregano derivatives and bioAgNP. Combinatory antimicrobial therapy is a potent strategy to control antimicrobial resistance, extend antimicrobial agents’ life, and also to reduce unwanted characteristics of compounds such as organoleptic features, toxicity, or costs [42,96,97].

Other studies have shown that bioAgNP presents an antimicrobial synergistic or additive effect when combined with different compounds, including essential oils or their main constituents [31,43,47,98,99,100,101,102,103,104,105,106]. Oregano-derived compounds also show an antimicrobial synergistic or additive effect when in combination [107,108,109,110,111,112]. In the previous study, our research group reported the synergistic and additive antibacterial interaction of oregano derivatives and bioAgNP (produced with *F. oxysporum*) against planktonic cells, including multidrug-resistant strains [31,47]. This present study shows, for the first time, the effect of four double-compound combinations composed of oregano terpenoids and this bioAgNP against bacteria living in biofilms.

All four combinations (OEO plus bioAgNP, Car plus bioAgNP, Thy plus bioAgNP, and Car plus Thy) inhibited the growth of biofilm, both at an early stage of formation and at the maturation phase, by EAEC and carbapenemase-producing *K. pneumoniae*. Our results are in agreement with the literature data, which show the antimicrobial potential of oregano-derived terpenoids and bioAgNP, both individually combined with conventional antimicrobials or natural compounds, to combat microbial biofilm [32,100,101]. In this present study, none of the four combinations showed antagonistic interaction with regard to antibiofilm activity since they were more efficient than antimicrobials individually by reducing SMIC_≥95_ of each compound (Table 2) or decreasing by greater intensity the biofilm biomass production and its viability in cases that SMIC_≥95_ reduction did not happen (Table 3, Table 4, Table 5 and Table 6).

For EAEC 042, Thy plus bioAgNP reduced SMIC_≥95_ against both biofilm formation and preformed biofilm compared to the same antimicrobials individually. Two combinations (OEO plus bioAgNP and Car plus bioAgNP) reduced SMIC_≥95_ against preformed biofilm; although both combinations did not reduce SMIC_≥95_ for biofilm formation, the combined compound still showed a better effect than isolated antimicrobials to prevent biofilm formation, since they caused significantly greater reduction in biofilm biomass and metabolic activity compared to antibacterials alone at same concentrations. Car plus Thy reduced SMIC_≥95_ against *E. coli* biofilm formation; both compounds in combination also showed an improved effect against preformed biofilm compared to antibacterials alone by reducing its viability in greater intensity.

For KPC-producing *K. pneumoniae* at an early stage of biofilm formation, three combinations (OEO plus bioAgNP, Car plus bioAgNP, and Thy plus bioAgNP) reduced SMIC_≥95_. Although the association containing Car and Thy did not cause a reduction in SMIC_≥95_ to prevent biofilm formation, both compounds, in combination, presented a better effect than both compounds alone, causing a greater reduction in biofilm biomass production and its viability.

For preformed biofilm of KPC strain, the SMIC_≥95_ of compounds in combination were not found; the maximum tested concentration of Thy or bioAgNP in combination was 25% of its individual SMIC value because both compounds alone at 0.5× SMIC already reduce around 80% of biofilm metabolic activity. However, two combinations, OEO plus bioAgNP and Car plus Thy, show greater antibiofilm activity than both compounds individually, causing a significantly greater reduction in sessile cell viability. Car plus bioAgNP and Thy plus bioAgNP showed similar antibiofilm activity compared to bioAgNP alone.

In general, this study shows that SMIC_≥95_ values reduce up to 50% for OEO and Car, 75% for Thy, and 87.5% for bioAgNP. These percentages of reduction are in agreement with the previous study of our research group in which MIC reduction against planktonic bacterial cells ranged by 50–87.5% for all compounds, showing the additive antibacterial effect of OEO plus bioAgNP, Car plus bioAgNP, Thy plus bioAgNP, and Car plus Thy [31,47].

Other authors also reported that oregano compounds (OEO or Car) combined with eugenol or conventional antibacterial (e.g., ciprofloxacin) present a synergistic effect to prevent and eradicate bacterial biofilms, including resistant strains [89,113]. Otaguiri et al. [100] showed that the same bioAgNP (produced extracellularly with *F. oxysporum* components) in combination with copaiba essential oil present a synergistic effect against *Streptococcus agalactiae* biofilm formation, reducing SMIC values of both compounds at least by 75%. This bioAgNP also showed an antibiofilm effect when combined with conventional antimicrobials. Longhi et al. [101] reported that their combination with fluconazole caused a significant decrease in the viability of both the initial and mature biofilm of *Candida albicans*.

At subinhibitory concentrations (lower than SMIC_≥95_), the four tested combinations showed antibiofilm activity against both EAEC and KPC-producing *K. pneumoniae* at the initial stage of biofilm formation (Table 3 and Table 4). At concentrations that do not inhibit planktonic cells (bellow SMIC_≥95_), OEO plus bioAgNP, Car plus bioAgNP, Thy plus bioAgNP, and Car plus Thy reduced total biofilm biomass and its metabolic activity compared to the untreated bacterium, suggesting that such treatments have an effect on specific pathways of biofilm lifestyle. Here we present microscopy and violacein assays as an initial study of the antibiofilm mechanism of these compounds alone and in combination.

SEM assay showed that compounds individually (Figure 2B–E, Figure 3B–E and Figure 4B–E), mainly Thy and bioAgNP, act against preformed EAEC biofilm on glass slides by affecting biofilm structure which presented reduced total biomass (extensively decreased cell density, less dense matrix, and less intercellular networks, with smaller and more scattered cellular aggregates) and its cells surface exhibited alterations. Both Thy and bioAgNP-treated biofilms presented cells with an irregular wrinkled surface; Thy also caused the sinking of the bacterial cell, and bioAgNP-treated cells also showed smaller sizes than typical *E. coli*. SEM analysis showed that OEO and Car alone affected EAEC biofilm to a lesser extent compared to other individual treatments. The OEO-treated sample presented slightly reduced biomass density and bacterial aggregation, whose cells exhibited a sinked and irregular wrinkled surface. The Car-treated sample showed biofilm with extremely poor intercellular networks, high cellular density, and cells without remarkable morphological alterations. Kerekes et al. [29] reported that Thy exhibited the best effect against *E. coli* biofilm among several essential oils, resulting in biofilm with anamorph structure, sparse micro-colonies, and individual cells (no aggregates), in agreement with our data. Guo et al. [92] reported the action of nanosilver against *P. aeruginosa* biofilm, which exhibited cellular density reduction and distinct EPS matrix formation surrounding bacterial cells with disruption of the cellular membrane.

The scanning microscopy test showed that two combinations (Car plus bioAgNP and Thy plus bioAgNP) stand out by inhibiting preformed EAEC biofilm growth on glass slides (Figure 2, Figure 3 and Figure 4; images G and H). Car plus bioAgNP and Thy plus bioAgNP samples presented damaged biofilm, with less dense biomass, matrix architecture with disrupted organization, a huge decrease in cell density, and smaller cellular aggregates, which shows loss of cell connections and morphological cell alterations such as reduced size and irregular wrinkled surface compared to untreated control. The OEO plus bioAgNP-treated sample showed less dense biofilm, with slightly less cellular density, fewer cell connections and bacterial aggregation, and altered cells with an irregular wrinkled surface. The Car plus Thy-treated sample showed biofilm with high cellular density without remarkable morphological alterations, but this treatment reduced intercellular networks. The cellular morphological alterations observed by us, such as irregular wrinkled surface and sinking of cellular surface, suggest that oregano derivatives and bioAgNP also affect sessile cells by disrupting the cytoplasmic membrane and cell wall, resulting in leakage of cellular cytoplasmic material in agreement with previous studies that involve planktonic cells [31,47,112,114,115].

In summary, we report in this paper the antibiofilm effect of new antimicrobial compositions against KPC-producing bacteria, which is even highlighted on the WHO Global Priority Pathogens List.Finally, this study also stands out the importance of using combined methodologies to access antibiofilm activity. Different methods trace the heterogeneity in biofilms and allow more reliable conclusions about the antimicrobial effect against sessile bacteria [51,54].

## 4. Materials and Methods

### 4.1. Bacterial Strains

Four bacterial strains were used in this study. Enteroaggregative *E. coli* (EAEC 042) and *E. coli* HB101 were used as standard strains for biofilm formation and negative control, respectively. KPC-producing *K. pneumoniae* (KPC-KP 52) from urinary tract infection was used as clinical isolate; it shows strong biofilm formation in vitro and presents a multidrug-resistant profile according to disk diffusion assay. KPC-KP 52 shows resistance to at least 15 antimicrobials (ampicillin, amoxicillin–clavulanate, cefazolin, cefepime, cefoxitin, ceftazidime, ceftazidime–clavulanate, aztreonam, ertapenem, gentamicin, ciprofloxacin, norfloxacin, nalidixic acid, chloramphenicol, and nitrofurantoin). *C. violaceum* CCT 3468 was used as a model for quorum-sensing inhibition assay. All bacterial samples were stored in Brain Heart Infusion (BHI, Acumedia, San Bernardino, CA, USA) broth containing 25% (*v*/*v*) glycerol (Merck, Rahway, NJ, USA) at −80 °C.

The KPC strain was provided by Dr. Mara Corrêa Lelles Nogueira (Faculdade de Medicina de São José do Rio Preto, São José do Rio Preto, São Paulo, Brazil). C. violaceum was provided by Dr. Marcelo Brocchi (Universidade Estadual de Campinas, Campinas, São Paulo, Brazil).

Both strains, EAEC 042 and KPC-producing *K. pneumoniae* strain, were chosen because they are great bacterial models for biofilm formation. EAEC 042 was used as a strong biofilm former (positive control; OD_570_ > 0.2 in crystal violet assay) and KPC-producing *K. pneumoniae* as a clinical isolate which is multidrug-resistant and also a strong biofilm former. The *E. coli* HB101 strain was used as a negative control that does not produce biofilm since it shows OD_570_ < 0.1.

### 4.2. Antibacterial Agents

#### 4.2.1. Oregano-Derived Compounds

OEO (batch 227) was obtained from Ferquima Industry and Commerce of Essential Oil (São Paulo, Brazil). It was extracted by steam distillation, and its main components (72% carvacrol, 2% thymol, 4.5% gamma-terpinene, 4% para-cymene, and 4% linalool) were described in a technical report provided by the company. Carvacrol-W224502 (Car) and thymol-T0501 (Thy) were purchased from Sigma-Aldrich (St. Louis, MO, USA); their densities are 0.95 g/mL and 0.976 g/mL, respectively. Individual solutions of OEO, Car, and Thy were prepared in dimethyl sulfoxide (DMSO, Sigma-Aldrich) for microbiological tests. DMSO maximum concentration in assays was 5% (*v/v*), and it did not show antibacterial action.

#### 4.2.2. Biogenically Synthetized Silver Nanoparticles (bioAgNP)

The bioAgNP synthesis was performed according to the previously established method [2]. The biosynthesis methodology involved *F. oxysporum* (strain 551 provided by ESALQ-USP Genetic and Molecular Biology Laboratory—Piracicaba, São Paulo, Brazil). Fungus was grown at 28 °C for 7 days in a medium composed of 0.5% (*w*/*v*) yeast extract (Becton, Dickinson and Company, Franklin Lakes, NJ, USA), 2% (*w*/*v*) agar malt extract (Acumedia), and distilled water. *F. oxysporum* biomass was added to distilled water at 0.1 g/mL and incubated at 28 °C for 72 h in agitation (150 rpm). Thereafter, aqueous solution components were separated from biomass by vacuum filtration (qualitative filter having an average pore size from 4 to 12 μm, Unifil). AgNO_3_ (Sigma-Aldrich) at 0.01 M was added to this solution, and it was kept at 28 °C for 15 days in the absence of light (static condition). The bioAgNP was obtained after the reduction of silver nitrate by fungal-free solution components. Aliquots of the system were removed for measuring absorption spectra to verify the surface plasmon resonance peak of bioAgNP, using ultraviolet-visible spectrophotometry (Thermo Scientific™ Multiskan™ GO Microplate Spectrophotometer, Marsiling Industrial Estate, Singapore). Washing of bioAgNP was carried out by three steps of centrifugation (27,000× *g*, 4 °C, 30 min) followed by incubation in an ultrasonic bath (30 min). Ag quantification was performed by Energy-Dispersive X-ray Fluorescence Spectrometer EDX-7000. The nanoparticle’s diameter was determined by photon-correlation spectroscopy using ZetaSizer NanoZS (Malvern, UK), and zeta-potential measurement was performed using the same instrument. Transmission electron microscopy (TEM, Zeiss EM900) was used for bioAgNP morphology analysis.

### 4.3. Antibiofilm Assays

The antibiofilm effect of antimicrobials alone and in combination was studied by four methodologies described above (Section 4.3.1, Section 4.3.2 and Section 4.3.3), such as colorimetric techniques (crystal violet and MTT) using microplates, scanning electron microscopy, and *quorum sensing* inhibition test.

#### 4.3.1. Biofilm Quantification by Chemical Methods (Crystal Violet and MTT)

The antibiofilm effect of oregano-derived compounds (OEO, Car, and Thy) and bioAgNP, individually and in combination, were evaluated at two stages of bacterial biofilm formation as follows: early stage (from 0 to 24 h of biofilm growth) and biofilm maturation phase (from 24 to 48 h). Both techniques were performed according to previously described methods, with necessary modifications, as follows: crystal violet test [116,117] and dimethylthiazol diphenyl tetrazolium bromide (MTT) assay [117,118].

For assessing the prevention of biofilm formation, antimicrobials and bacteria were added concomitantly to the microtiter plate. Briefly, bacterial-isolated colonies grown in nutrient-agar (Himedia, Mumbai, India) medium were suspended in phosphate-buffered saline (0.1 M PBS, pH 7.2) to standardize the inoculum density. This suspension was adjusted to achieve turbidity equivalent to 0.5 McFarland standard, which corresponds approximately to 1.5 × 10^8^ colony-forming units (CFU)/mL. Bacteria and antimicrobials were added concomitantly to wells of 96-well polystyrene microtiter plate; a volume of 0.02 mL of equivalent 0.5 McFarland suspension was added to 0.18 mL Dulbecco’s Modified Eagle’s Medium (DMEM, Sigma-Aldrich) containing antimicrobials individually or in combination. Before adding antimicrobials, DMEM was supplemented with 0.45% glucose (Sigma-Aldrich). The plate was incubated at 37 °C for 24 h under static conditions to allow bacterial form biofilm. Two identical microtiter plates were prepared; one for the crystal violet-staining procedure and the other for the MTT assay.

For evaluating the effect of compounds against preformed biofilm, firstly, non-treated bacteria were added to a microtiter plate and incubated for 24 h to allow biofilm attachment and growth, then preformed biofilm was treated with antimicrobials. Briefly, the bacterial inoculum was prepared as previously described; 0.02 mL of equivalent 0.5 McFarland suspension and DMEM supplemented with 0.45% glucose (0.18 mL) were added to wells of a 96-well polystyrene microtiter plate, followed by incubation at 37 °C for 24 h. Thereafter, unattached cells and medium were removed, and biofilm biomass was rinsed three times with PBS. Then 0.2 mL of DMEM alone (untreated control) or DMEM containing antimicrobials (individually and in combination) were added to preformed biofilm, followed by post-incubation at 37° for 24 h.

After 24 h of treatment (for both biofilm at the initial stage and preformed biofilm), planktonic cells and DMEM were aspired off, and adherent biomass was rinsed three times with PBS. For the crystal violet assay, biomass was stained with 0.2% (*w*/*v*) crystal violet solution for five min; then, three washing steps were carried out to remove unbound dye. Finally, after adding 0.2 mL of ethanol 95% (*v*/*v*) to each well containing the bound dye, the biofilm was quantified by a microplate reader at 570 nm (Thermo Scientific™ Multiskan™ GO Microplate Spectrophotometer). For viability assay, MTT solution (0.1 mL per well at 0.25 mg/mL) was added to each well, and the microplate was incubated at 37 °C for 2 h. Thereafter, 0.1 mL of solubilization solution was added to each well to dissolve formazan crystals. After 15 min homogenization, the plate was read at 570 nm using the same microplate spectrophotometer.

Untreated bacteria, inoculated on DMEM alone or containing DMSO at 5% (*v*/*v*), were used as a positive control (PC, defined as 100% biofilm metabolic activity). DMEM alone was used as sterility control (SC). The percentage of biofilm inhibition (total biomass and metabolic activity reductions) for each antimicrobial treatment was calculated using Equation (1).
(1)$$Biofilminhibition\;(\%)=100-\left[\frac{({OD}_{570}\;treatment-{OD}_{570}\;SC)\times 100}{{OD}_{570}\;PC-{OD}_{570}\;SC}\right]$$

The sessile (biofilm) minimum inhibitory concentrations were determined at 95% or greater inhibition (SMIC_≥95_) of metabolic activity compared to untreated positive control. Experiments were carried out in quintuplicate on at least three different occasions. Details of methodologies are described below.

Four antimicrobials were tested individually (OEO, Car, Thy, and bioAgNP), whose concentrations ranged as follows: (i) 0.07–9.5 mg/mL for OEO, (ii) 0.08–9.76 mg/mL for Car, (iii) 0.01–2 mg/mL for Thy, and (iv) 0.09–740 µM for bioAgNP. Four double-antimicrobial combinations were tested (OEO plus bioAgNP, Car plus bioAgNP, Thy plus bioAgNP, and Car plus Thy), whose final concentrations ranged as follows: (i) 0.02–0.3 mg/mL for OEO, (ii) 0.02–0.31 mg/mL for Car, (iii) 0.008–0.25 mg/mL for Thy, and (iv) 0.03–7.88 µg/mL for bioAgNP.

PBS 0.1 M (pH 7.2) was composed of 0.9% (*w*/*v*) NaCl, 0.2 M monobasic sodium phosphate (Chemco), and 0.2 M dibasic sodium phosphate (Nuclear). The solubilization solution was composed of isopropanol containing triton X-100 at 1% (*v*/*v*) and HCl at 0.36% (*v/v*) (1 N HCl was used to prepare the final solubilization solution).

#### 4.3.2. Quorum Sensing Inhibition Test Based on *C. violaceum*

*C. violaceum* CCT 3468 was used as a model for quorum sensing inhibition assay since violacein production involves quorum sensing. Before quantitative analysis of violacein production, the minimum inhibitory concentration (MIC) of compounds (individually and in combination) and their subinhibitory concentrations were determined. In addition, quantification of viable bacterial cells was performed for treated and untreated *C. violaceum*. Details of the violacein assay are described below.

##### Determination of Subinhibitory Antibacterial Concentrations

Before the violacein inhibition assay, subinhibitory concentrations of each compound against *C. violaceum* were determined by the broth microdilution method. Determination of MIC of each antimicrobial individually (OEO, Car, Thy, and bioAgNP) was performed according to the Clinical and Laboratory Standards Institute guidelines [119], with necessary modifications. For antimicrobial combination (Thy plus bioAgNP), MIC values were determined by double-antimicrobial gradient as described by Traub and Kleber [120], with necessary modifications. Briefly, to standardize the inoculum density for the susceptibility test, *C. violaceum* isolated colonies grown in Luria Bertani (LB, Himedia) agar medium were suspended in PBS 0.1M (pH 7.2) to achieve turbidity equivalent to 0.5 McFarland standard, as previously described for microtiter assays. The equivalent 0.5 McFarland suspension was diluted 1:100 in LB (Himedia) broth to obtain a concentration of approximately 10^6^ CFU/mL. A volume of 0.05 mL of bacterial inoculum at 10^6^ CFU/mL was added to 0.05 mL of LB containing antimicrobial individually or in combination. Lastly, bacteria at 5 × 10^5^ CFU/mL in LB containing antimicrobials were incubated at 28 °C for 24 h with shaking (130 rpm). MIC was defined as the lowest antimicrobial concentration that inhibited visible growth after 24 h of treatment at 28 °C. The assay was carried out in triplicate, at least on three different occasions.

For antimicrobials tested individually, concentrations ranged as follows: (i) 0.07–9.5 mg/mL for OEO, (ii) 0.08–9.76 mg/mL for Car, (iii) 0.008–1 mg/mL for Thy, and (iv) 0.49–63 µg/mL for bioAgNP. For the combination assay, the concentration range was 0.01–0.06 mg/mL for Thy and 0.49–1.97 µg/mL for bioAgNP. LB alone and LB containing each antimicrobial separately were tested as sterility controls. Untreated bacteria inoculated on LB broth alone and containing DMSO at 5% (*v*/*v*) were tested as growth control.

##### Violacein Inhibition Assay

At first, quantification of viable cells in *C. violaceum*-treated samples was performed. *C. violaceum* was grown in LB broth at 28 °C for 72 h (130 rpm); every 24 h, the medium was renewed by transference of 0.1 mL of each previous culture into LB broth (4.9 mL). For the violacein inhibition assay, *C. violaceum* overnight culture was diluted 1:10 in LB broth; then six samples were prepared by adding 2.5 mL of diluted bacterial inoculum to 2.5 mL of LB alone (untreated control) or LB containing antimicrobial (subinhibitory concentrations) individually or in combination, whose concentrations were as follows: (i) OEO at 0.15 mg/mL, (ii) Car at 0.15 mg/mL, (iii) Thy at 0.12 mg/mL, (iv) bioAgNP at 15.75 µg/mL, and (v) combination with Thy at 0.03 mg/mL plus bioAgNP at 0.49 µg/mL. The untreated and treated bacterial samples were incubated at 28 °C for 24 h (130 rpm). After 24 h treatment, each sample was evaluated with regard to the number of *C. violaceum* viable cells according to the National Committee for Clinical Laboratory Standards [121]; 0.01 mL from serial dilutions (in PBS 0.1M, pH 7.2) of treated and non-treated cultures were subcultured in LB agar for CFU/mL determination.

The amount of violacein produced by each sample (treated and non-treated *C. violaceum*) was qualitatively analyzed (turbidity and color of bacterial cultures were analyzed by visual inspection) according to Blosser and Gray [122]. Briefly, bacterial cells of each sample were pelleted (5500× *g*, 10 min, 25 °C) and resuspended with 0.2 mL 0.1 M PBS (pH 7.2). Bacterial cells were lysed by adding 0.2 mL of 10% (*w*/*v*) sodium dodecyl sulfate (SDS), mixing for 10 s with a vortex mixer, and cells were maintained at room temperature for 5 min. For violacein extraction, 0.9 mL of water-saturated butanol (1:3) was added to cell lysate, followed by mixing for 5 s. The final solution was centrifuged (13,000× *g*, 5 min). The upper n-butanol phase containing violacein was collected and transferred to a 96-wells plate. The absorbance of extracted violacein was measured at 595 nm in a microplate reader (Thermo Scientific™ Multiskan™ GO Microplate Spectrophotometer). Untreated bacteria inoculated on LB broth alone or containing DMSO at 5% (*v*/*v*) were used as a positive control (defined as 100% of violacein production). The percentage of violacein produced by treated cells was calculated using Equation (2). For each sample, the percentage of inhibition of violacein production was determined based on the positive control, subtracting the percentage of violacein from 100. The assay was carried out in triplicate, at least on three different occasions.
(2)$$Violaceinproduction\;(\%)=\left[\frac{{OD}_{595}\;treatment\times 100}{{OD}_{595}\;untreated\;control}\right]$$

#### 4.3.3. Scanning Electron Microscopy (SEM) Study of Antibiofilm Effect of Compounds

The effect of oregano-derived compounds (OEO, Car, and Thy) and bioAgNP, individually and in combination, against the preformed biofilm of EAEC 042 was analyzed by SEM. Firstly, the bacterial inoculum was prepared as described in Section 4.3.1. Bacteria (0.1 mL of equivalent 0.5 McFarland suspension) and DMEM supplemented with 0.45% glucose (0.9 mL) were added to wells (which contained uncoated glass slides at the bottom) of a 24-well polystyrene microtiter plate, followed by incubation at 37 °C for 24 h in agitation (120 rpm) to allow cell attachment and biofilm growth. Thereafter, unattached cells and medium were removed, and biofilm biomass was rinsed three times with PBS 0.1 M (pH 7.2). Then 0.2 mL of DMEM alone (untreated control) or DMEM containing antimicrobials (individually and in combination) were added to the preformed biofilm, followed by incubation at 37 °C for 24 h (120 rpm).

As biofilm was formed on a different surface (glass slides), the antibacterial concentrations were not chosen based on a 96-well polystyrene microplate assay. Therefore, SMIC_100_ of treatments was determined under specific conditions for SEM assay using a crystal violet test. Thus, eight treatments at 0.5 × SMIC_100_ (sessile minimum-inhibitory concentration which eliminated 100% of preformed biofilm biomass) were evaluated by electron microscopy as follows: (1) OEO at 0.3 mg/mL, (2) Car at 0.31 mg/mL, (3) Thy at 0.12 mg/mL, (4) bioAgNP at 7.88 µg/mL, (5) OEO (0.15 mg/mL) plus bioAgNP (3.94 µg/mL), (6) Car (0.15 mg/mL) plus bioAgNP (3.94 µg/mL), (7) Thy (0.06 mg/mL) plus bioAgNP (3.94 µg/mL), and (8) Car (0.15 mg/mL) plus Thy (0.06 mg/mL).

After 24 h treatment, planktonic cells and DMEM were aspired off, adherent biomass was rinsed three times with PBS 0.1M (pH 7.2), and preparation of samples (treated and untreated bacteria) for SEM analyses was performed accordingly [123], with necessary modifications. Previously, four solutions for microbial glycocalyx fixation were tested (data not shown), and the solution containing alcian blue as a cationic dye was chosen as the most suitable one for EAEC 042. Firstly, glass slides with adherent biomass (treated and untreated samples) were immersed for 20 h (at 4 °C) in 1 mL of 0.1M sodium cacodylate buffer (pH 7.2) containing 2.5% (*v*/*v*) glutaraldehyde, 2% (*v*/*v*) paraformaldehyde, and 0.15% (*w*/*v*) alcian blue. After primary fixation in aldehyde with alcian blue, the samples were washed (three washing steps of 10 min each) in 0.1 M cacodylate buffer (pH 7.2), following post-fixation in OsO_4_ 1% for 2 h at room temperature. All reagents for both chemical fixations were provided by Electron Microscopy Sciences. Post-fixed samples were then rinsed (three times for 10 min each) in 0.1 M cacodylate buffer (pH 7.2) and dehydrated in an ethanol gradient (Sigma-Aldrich) (30, 50, 70, 90, and 100 °GL), critical point-dried using CO_2_ (BALTEC CPD 030 Critical Point Dryer), coated with gold (BALTEC SDC 050 Sputter Coater) and observed under scanning electron microscope (FEI Quanta 200).

## 5. Conclusions

This study shows for the first time the antibiofilm effect of bioAgNP combined with oregano compounds (OEO, Car, or Thy) against *E. coli* and *K. pneumoniae*, including KPC-producing strains for which new antibiotics are urgently needed. Binary-compound combinations improved the antibiofilm effect of antimicrobials alone, disrupting preformed biofilm and preventing its formation. We highlight the great antibacterial activity of Thy associated with bioAgNP, which inhibited the growth of biofilm on both polystyrene and glass surfaces, reduced SMIC_≥95_ of each compound, decreased biofilm metabolic activity and biomass, disrupted its three-dimensional structure, and altered its cell morphology; Thy plus bioAgNP also reduced violacein production by *C. violaceum*, indicating that disruption of quorum sensing may be one of its antibiofilm mechanisms. Next, a more detailed examination of oregano plus bioAgNP must be performed to provide information with regard to the antibiofilm mechanism of action (at a molecular level) and its antibiofilm efficacy in vivo and in non-laboratory situations. However, terpenoids derived from oregano associated with bioAgNP (synthesized with *F. oxysporum*) successfully combat biofilm-associated bacteria and may overcome existing antibiotic resistance so they could be applied in several sectors of industry, clinical, and hospital settings, such as formulation of surface cleaners, food packaging, cosmetic products, wound care supplies, for treating infection in burns, among others.

## Figures and Tables

**Figure 1 antibiotics-12-00756-f001:**
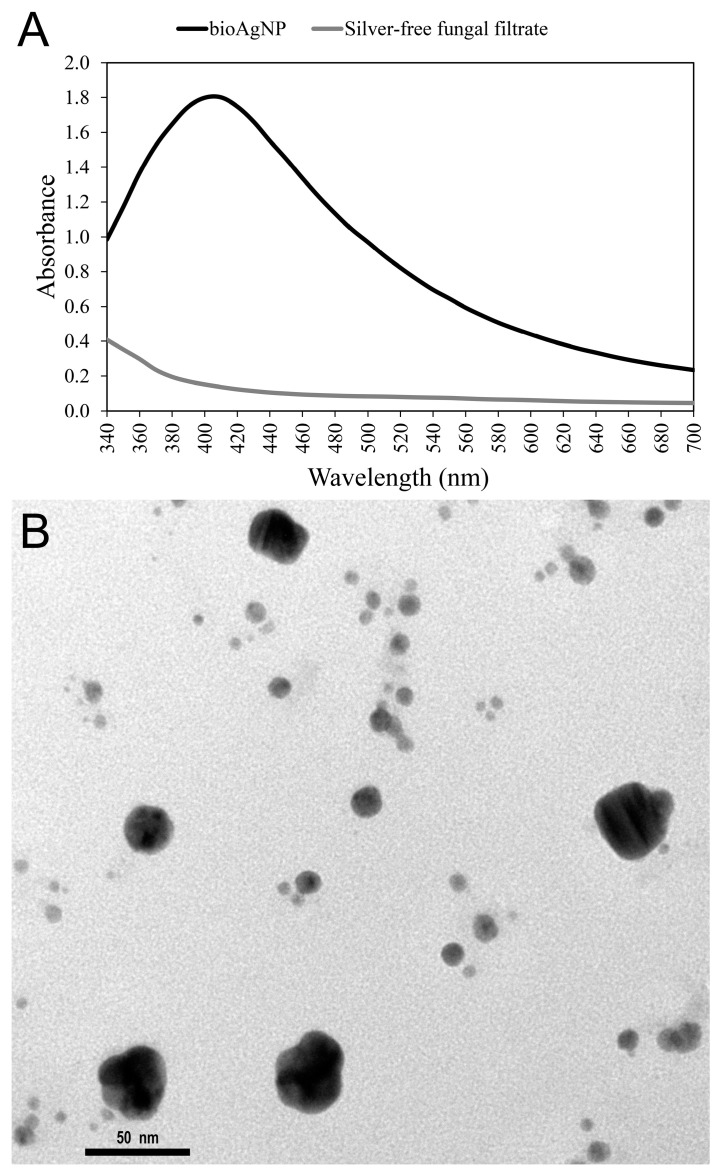
Characterization of biogenic silver nanoparticles (bioAgNP). (**A**) Plasmonic band: UV-Vis spectra show that the plasmon resonance of bioAgNP is centered at 420 nm, and this absorption peak is not observed for a fungal-free solution (negative control). (**B**) Morphology: MET micrograph of bioAgNP shows spherical nanoparticles. The size and zeta distributions of bioAgNP and calibration curve used to determine silver concentration are shown in Appendix A.

**Figure 2 antibiotics-12-00756-f002:**
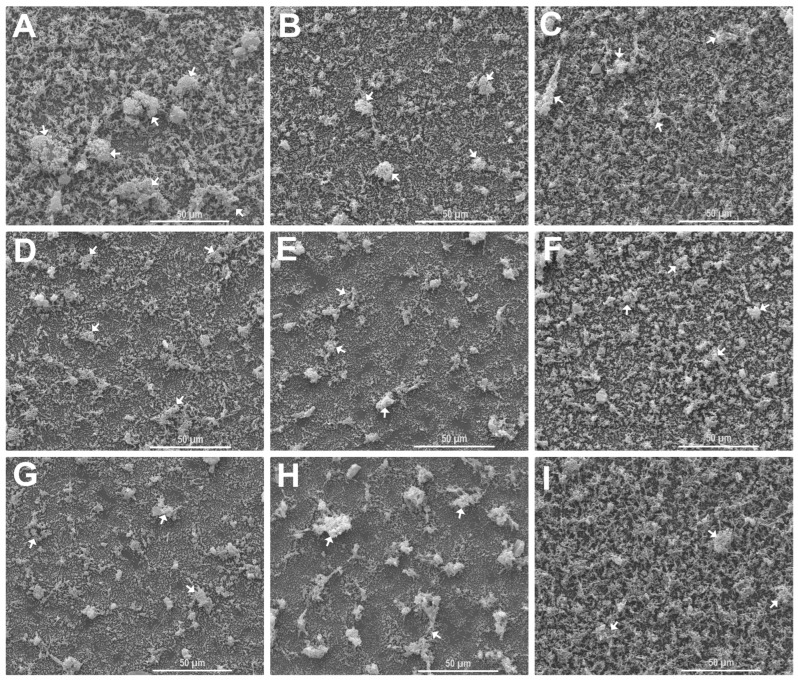
Low magnification (1600×) scanning electron micrographs of the antibiofilm effect of oregano-derived antimicrobials and bioAgNP individually and in combination against enteroaggregative *Escherichia coli* (EAEC 042), showing the amount of biofilm on glass surface seen as cell density and cell aggregates. Preformed biofilms (24 h) in glass slides were exposed for 24 h to eight different treatments at subinhibitory concentrations. (**A**) Untreated control (biofilm at a later stage, 48 h of growth). (**B**) OEO at 0.3 mg/mL. (**C**) Car at 0.31 mg/mL. (**D**) Thy at 0.12 mg/mL. (**E**) bioAgNP at 7.88 µg/mL. (**F**) Combination of OEO (0.01 mg/mL) plus bioAgNP (3.94 µg/mL). (**G**) Combination of Car (0.15 mg/mL) plus bioAgNP (3.94 µg/mL). (**H**) Combination of Thy (0.06 mg/mL) plus bioAgNP (3.94 µg/mL). (**I**) Combination of Car (0.15 mg/mL) plus Thy (0.06 mg/mL). Micrographs (**A**–**I**) show cell density and biofilm cell clusters. Arrows: cell aggregates.

**Figure 3 antibiotics-12-00756-f003:**
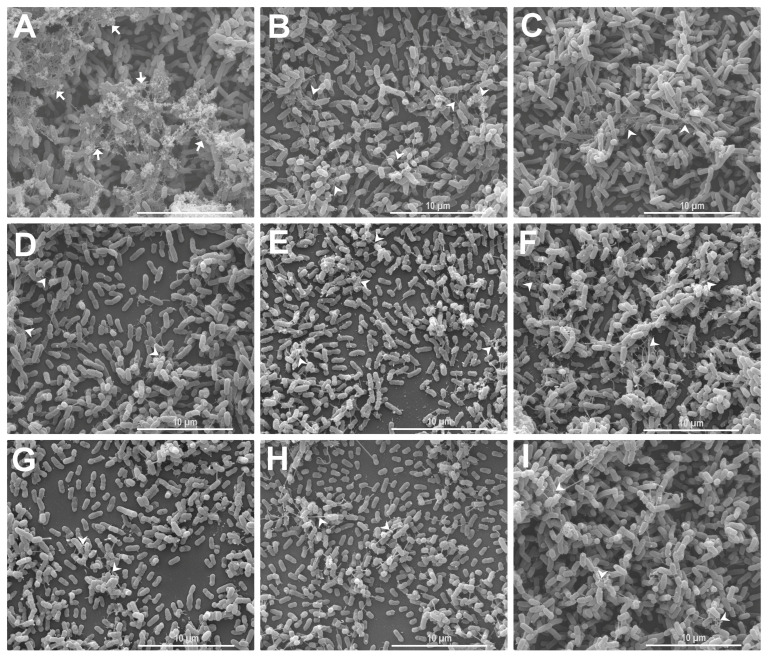
Scanning electron micrographs (magnification 10,000×) of the antibiofilm effect of oregano-derived antimicrobials and bioAgNP individually and in combination against enteroaggregative *Escherichia coli* (EAEC 042), showing biofilm structure. Preformed biofilms (24 h) in glass slides were exposed for 24 h to eight different treatments at subinhibitory concentrations. (**A**) Untreated control (biofilm at a later stage, 48 h of growth). (**B**) OEO at 0.3 mg/mL. (**C**) Car at 0.31 mg/mL. (**D**) Thy at 0.12 mg/mL. (**E**) bioAgNP at 7.88 µg/mL. (**F**) Combination of OEO (0.01 mg/mL) plus bioAgNP (3.94 µg/mL). (**G**) Combination of Car (0.15 mg/mL) plus bioAgNP (3.94 µg/mL). (**H**) Combination of Thy (0.06 mg/mL) plus bioAgNP (3.94 µg/mL). (**I**) Combination of Car (0.15 mg/mL) plus Thy (0.06 mg/mL). Micrographs (**A**–**I**) show the cell density and exopolysaccharide matrix of EAEC 042 in detail, including the unstructured matrix. Arrows: structured exopolysaccharide matrix with undamaged intercellular networks. Arrowheads: unstructured exopolysaccharide matrix with poor intercellular networks.

**Figure 4 antibiotics-12-00756-f004:**
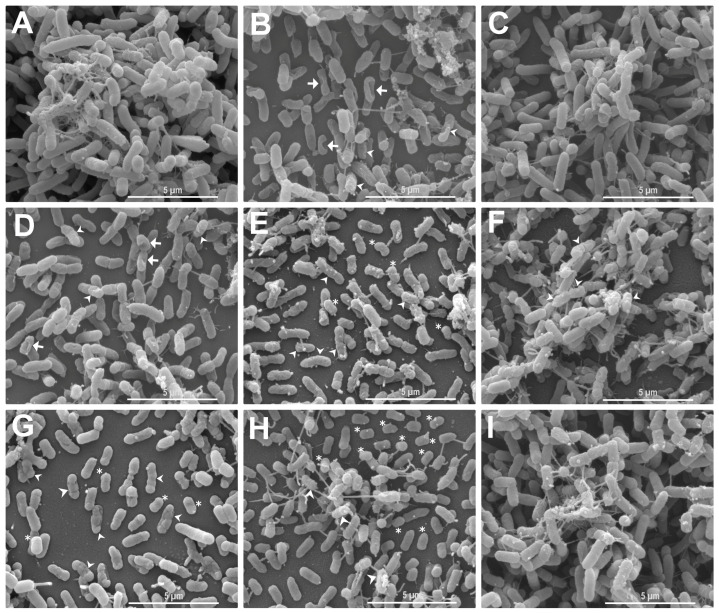
Scanning electron micrographs (magnification 20,000×) of the antibiofilm effect of oregano-derived antimicrobials and bioAgNP individually and in combination against enteroaggregative *Escherichia coli* (EAEC 042), showing biofilm cellular alterations in detail. Preformed biofilms (24 h) in glass slides were exposed for 24 h to eight different treatments at subinhibitory concentrations. (**A**) Untreated control (biofilm at a later stage, 48 h of growth). (**B**) OEO at 0.3 mg/mL. (**C**) Car at 0.31 mg/mL. (**D**) Thy at 0.12 mg/mL. (**E**) bioAgNP at 7.88 µg/mL. (**F**) Combination of OEO (0.01 mg/mL) plus bioAgNP (3.94 µg/mL). (**G**) Combination of Car (0.15 mg/mL) plus bioAgNP (3.94 µg/mL). (**H**) Combination of Thy (0.06 mg/mL) plus bioAgNP (3.94 µg/mL). (**I**) Combination of Car (0.15 mg/mL) plus Thy (0.06 mg/mL). Micrographs (**A**–**I**) show the cell density, size, and shape of EAEC 042 and morphological changes on the cell surface. Arrows: sinking of the bacterial cell wall. Arrowheads: wrinkled cell surface. Asterisk: cells with reduced size compared to untreated control.

**Figure 5 antibiotics-12-00756-f005:**
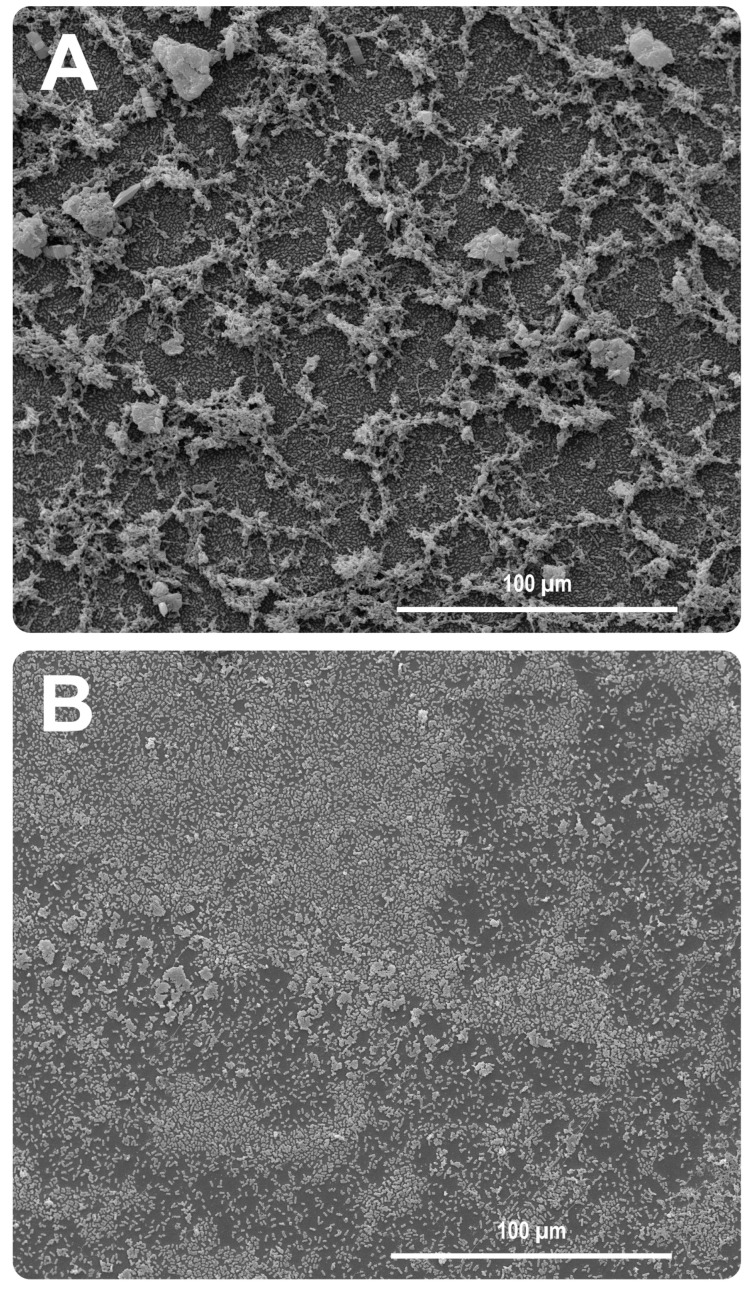
Scanning electron micrographs of low magnification (1000×) of enteroaggregative *Escherichia coli* (EAEC 042) and KPC-producing *K. pneumoniae* biofilms at an early stage of formation. Both biofilms were grown on a glass surface for 24 h. (**A**) EAEC biofilm shows the initial development of three-dimensional architecture, with cellular aggregates and a remarkable amount of EPS. (**B**) KPC biofilm in the initial stage of development, without cell aggregates, no exopolysaccharide matrix, and absence of three-dimensional structure.

**Figure 6 antibiotics-12-00756-f006:**
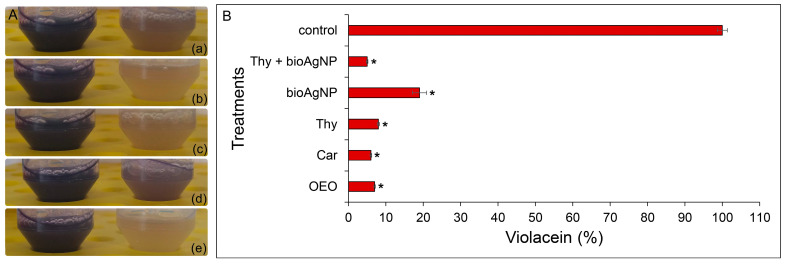
Effect of oregano-derived antimicrobials and bioAgNP individually and in combination on violacein production by *Chromobacterium violaceum*. *C. violaceum* was exposed for 24 h to five treatments at subinhibitory concentrations as follows: OEO at 0.07 mg/mL, Car at 0.04 mg/mL, Thy at 0.06 mg/mL, bioAgNP at 3.94 µg/mL, and combination composed of Thy at 0.008 mg/mL and bioAgNP at 0.49 µg/mL. (**A**) It shows the result of a qualitative analysis of the effect of antimicrobials on violacein production by *C. violaceum*. The first tube of each pair is untreated control, which shows high production of violacein pigment after 24 h of incubation at 28 °C. The second tube of each pair is *C. violaceum* treated with antibacterials at subinhibitory concentrations for 24 h as follows: (**a**) OEO-treated bacterial cells, (**b**) Car-treated bacterial sample, (**c**) Thy-treated bacterial sample, (**d**) bioAgNP-treated bacterial cells, and (**e**) bacterial cells treated with a combination of Thy plus bioAgNP. The color intensity of violet indicates the amount of violacein pigment. The turbidity of the broth medium indicates that compounds reduced violacein production without affecting bacterial growth. (**B**) It shows a quantitative analysis of the amount of violacein produced by *C. violaceum* treated with antimicrobials individually and in combination. Extracted violacein was measured at 595 nm. The amount of pigment produced by each treated bacterial sample is based on the amount produced by the untreated sample (control which produces 100% of pigment as determined). Amounts of violacein (%) are the mean ± standard deviation.* Indicates a statistically significant difference (*p* < 0.05, Kruskal–Wallis test) between treated and untreated control in terms of violacein production.

**Table 1 antibiotics-12-00756-t001:** Sessile minimal inhibitory concentration (SMIC_≥95_) of oregano derivatives (mg/mL) and biological silver nanoparticles (bioAgNP, µg/mL) individually against enteroaggregative *Escherichia coli* (EAEC 042) and KPC-producing *Klebsiella pneumoniae* (KPC), at both early stage of biofilm formation and preformed biofilm.

Antimicrobial	Formation	Preformed	Formation	Preformed
OEO	0.30	0.59	0.59	0.59
Car	0.31	0.61	0.31	0.31
Thy	0.25	0.5	0.25	0.25
bioAgNP	0.98	31.5	1.97	7.88

SMIC_≥95_: Antimicrobial concentration, which resulted in at least a 95% reduction in metabolic activity of sessile cells. SMIC_≥95_ values were determined according to the results of the MTT assay. For biofilm formation, all SMIC_≥95_ cause a 100% reduction in biofilm viability. Oregano derivatives: OEO (oregano essential oil), Car (carvacrol), and Thy (thymol).

**Table 2 antibiotics-12-00756-t002:** Sessile minimal inhibitory concentration (SMIC_≥95_) of oregano derivatives (mg/mL) and biological silver nanoparticles (bioAgNP, µg/mL) in binary combinations against enteroaggregative *Escherichia coli* (EAEC 042) and KPC-producing *Klebsiella pneumoniae* (KPC), at both early stage of biofilm formation and preformed biofilm.

Antibacterial Binary Combinations	EAEC				KPC			
	Formation		Preformed		Formation		Preformed	
	SMIC Combination	Fold Decrease	SMIC Combination	Fold Decrease	SMIC Combination	Fold Decrease	SMIC Combination	Fold Decrease
**OEO + bioAgNP**								
OEO	>0.15	no	0.30	2×	0.30	2×	>0.30	no
bioAgNP	>0.25	NT	7.88	4×	0.98	2×	>1.97	NT
**Car + bioAgNP**								
Car	>0.15	no	0.31	2×	0.15	2×	>0.15	no
bioAgNP	>0.25	NT	7.88	4×	0.98	2×	>1.97	NT
**Thy + bioAgNP**								
Thy	0.12	2×	0.12	4×	0.12	2×	>0.06	NT
bioAgNP	0.25	4×	3.94	8×	0.98	2×	>1.97	NT
**Car + Thy**								
Car	0.15	2×	>0.31	no	>0.15	no	>0.15	no
Thy	0.12	2×	>0.25	no	>0.12	no	>0.06	NT

Oregano derivatives: OEO (oregano essential oil), Car (carvacrol), and Thy (thymol). SMIC_≥95_: Antimicrobial concentration, which resulted in at least a 95% reduction in metabolic activity of sessile cells. Fold decrease describes how much the SMIC of both compounds in combination was reduced in comparison to the SMIC of the same compounds individually (SMIC values of compounds alone are shown in Table 1). NT (not tested): The maximum tested concentration of Thy or bioAgNP in combination was 25% of their individual SMIC values because both compounds alone at 0.5 × SMIC already reduce more than 80% of biofilm metabolic activity. However, both Thy and bioAgNP in combinations show greater antibiofilm activity than both compounds individually, as shown in Table 3 and Table 4.

**Table 3 antibiotics-12-00756-t003:** Antibiofilm effect, shown in terms of biomass and metabolic activity reduction, of oregano derivatives combined with biological silver nanoparticles (bioAgNP) compared to both antimicrobials individually against enteroaggregative *Escherichia coli* (EAEC 042) biofilms growth in microtiter plates, which were evaluated at an early stage of biofilm formation.

Bacteria	Antimicrobial Concentrations in Binary Combinations	Biofilm Reduction Caused by Combinations		Antibiofilm Effect of Combinations	Antimicrobial Concentrations Individually	Biofilm Reduction Caused by Antibacterials Alone	
		Biomass Decrease (%)	Metabolic Activity Decrease (%)			Biomass Decrease (%)	Metabolic Activity Decrease (%)
**EAEC**	**OEO + bioAgNP**			Improved	OEO at 0.15mg/mL	40 ± 1.9	3 ± 2.8
	0.15mg/mL + 0.25 µg/mL	88 ± 0.5 *	70 ± 2.9 *		OEO at 0.07mg/mL	19 ± 0.9	0 ± 0.0
	0.07 mg/mL + 0.12 µg/mL	49 ± 2.6 *	6 ± 0.3		Car at 0.15 mg/mL	60 ± 2.8	2 ± 2.1
	**Car + bioAgNP**			Improved	Car at 0.08 mg/mL	14± 1.8	0 ± 0.0
	0.15 mg/mL + 0.25 µg/mL	84 ± 0.7 *	64 ± 1.5 *		Thy at 0.12 mg/mL	0 ± 0.0	0 ± 0.0
	0.08 mg/mL + 0.12 µg/mL	51 ± 1.5 *	9 ± 0.7		Thy at 0.06 mg/mL	0 ± 0.0	0 ± 0.0
	**Thy + bioAgNP**			Improved	bioAgNP at 0.25 µg/mL	66 ± 1.2	27 ± 1.2
	0.12 mg/mL + 0.25 µg/mL	99 ± 0.6 *	98 ± 0.1 *		bioAgNP at 0.12 µg/mL	21 ± 1.4	13 ± 0.8
	0.06 mg/mL + 0.12 µg/mL	22 ± 1.4	9 ± 0.2				
	**Car + Thy**			Improved			
	0.15 mg/mL + 0.12 mg/mL	99 ± 0.5 *	99 ± 0.1 *				
	0.08 mg/mL + 0.06 mg/mL	62 ± 0.9 *	19 ± 1.3 *				

Oregano derivatives: OEO (oregano essential oil), Car (carvacrol), and Thy (thymol). * It indicates that the binary combination caused statistically (*p* < 0.05, Kruskal–Wallis test) a greater reduction in biofilm formation and showed an improved antibiofilm effect compared to both antimicrobials alone at the same concentrations. When the combination has an antibacterial effect similar to the antimicrobials alone, the difference in biofilm reduction is not significant. ±(standard deviation).

**Table 4 antibiotics-12-00756-t004:** Antibiofilm effect, shown in terms of biomass and metabolic activity reduction, of oregano derivatives combined with biological silver nanoparticles (bioAgNP) compared to both antimicrobials individually against KPC-producing *Klebsiella pneumoniae* (KPC) biofilms growth in microtiter plates, which were evaluated at an early stage of biofilm formation.

Bacteria	Antimicrobial Concentrations in Binary Combinations	Biofilm Reduction Caused by Combinations		Antibiofilm Effect of Combinations	Antimicrobial Concentrations Individually	Biofilm Reduction Caused by Antibacterials Alone	
		Biomass Decrease (%)	Metabolic Activity Decrease (%)			Biomass Decrease (%)	Metabolic Activity Decrease (%)
**KPC**	**OEO + bioAgNP**			Improved	OEO at 0.30mg/mL	54 ± 1.7	42 ± 1.1
	0.30 mg/mL+ 0.98 µg/mL	97 ± 0.8 *	99 ± 0.1 *		OEO at 0.15 mg/mL	43 ± 1.9	33 ± 0.3
	0.15 mg/mL + 0.49µg/mL	54 ± 0.9	58 ± 2.1 *		Car at 0.15 µg/mL	23 ± 0.3	24 ± 2.5
	**Car + bioAgNP**			Improved	Car at 0.08 µg/mL	15 ± 1.9	27 ± 1.8
	0.15 mg/mL + 0.98µg/mL	97 ± 0.1 *	99 ± 0.2 *		Thy at 0.12 µg/mL	20 ± 3.1	6 ± 0.1
	0.08 mg/mL + 049 µg/mL	59 ± 1.7 *	21 ± 2.6		Thy at 0.06 µg/mL	2 ± 0.7	0 ± 0.0
	**Thy + bioAgNP**			Improved	bioAgNP at 0.98 µg/mL	0 ± 0.00	0 ± 0.0
	0.12 mg/mL + 0.98 µg/mL	99 ± 0.4 *	100 ± 0.0 *		bioAgNP at 0.49 µg/mL	0 ± 0.00	0 ± 0.0
	0.06 mg/mL + 0.49 µg/mL	12 ± 0.5	0 ± 0.0				
	**Car + Thy**			Similar			
	0.15 mg/mL + 0.12 mg/mL	34 ± 1.3	34 ± 2.9				
	0.08 mg/mL + 0.06 mg/mL	19 ± 1.6	8 ± 1.7				

Oregano derivatives: OEO (oregano essential oil), Car (carvacrol), and Thy (thymol). * It indicates that the binary combination caused statistically (*p* < 0.05, Kruskal–Wallis test) a greater reduction in biofilm formation and showed an improved antibiofilm effect compared to both antimicrobials alone at the same concentrations. When the combination has an antibacterial effect similar to the antimicrobials alone, the difference in biofilm reduction is not significant. ±(standard deviation).

**Table 5 antibiotics-12-00756-t005:** Antibiofilm effect, which is shown in terms of metabolic activity reduction of oregano derivatives and biological silver nanoparticles (bioAgNP) compared to both antimicrobials individually against enteroaggregative *Escherichia coli* (EAEC 042) biofilms growth in microtiter plates, which were evaluated at preformed biofilm condition.

Bacteria	Antimicrobial Concentrations in Binary Combinations	Biofilm Reduction Caused by Combinations	Antibiofilm Effect of Combinations Compared to Antimicrobials Alone	Antimicrobial Concentrations Individually	Biofilm Reduction Caused by Antibacterials Alone
		Metabolic Activity Decrease (%)			Metabolic Activity Decrease (%)
**EAEC**	**OEO + bioAgNP**		Improved	OEO at 0.30 mg/mL	19 ± 0.3
	0.30 mg/mL + 7.88 µg/mL	99 ± 0.1 *		OEO at 0.15 mg/mL	5 ± 0.9
	0.15 mg/mL + 3.94 µg/mL	41 ± 2.1 *		Car at 0.31 mg/mL	12 ± 3.1
	**Car + bioAgNP**		Improved	Car at 0.15 mg/mL	7 ± 0.6
	0.31 mg/mL + 7.88 µg/mL	99 ± 0.1 *		Thy at 0.25 mg/mL	13 ± 1.2
	0.15 mg/mL + 3.94 µg/mL	26 ± 0.2		Thy at 0.12 mg/mL	1 ± 0.9
	**Thy + bioAgNP**		Improved	bioAgNP at 7.88 µg/mL	70 ± 2.8
	0.25 mg/mL + 7.88 µg/mL	100 ± 0.0 *		bioAgNP at 3.94 µg/mL	28 ± 1.8
	0.12 mg/mL + 3.94 µg/mL	98 ± 0.5 *			
	**Car + Thy**		Improved		
	0.31 mg/mL + 0.25 mg/mL	93 ± 0.2 *			
	0.15 mg/mL + 0.12 mg/mL	92 ± 0.3 *			

Oregano derivatives: OEO (oregano essential oil), Car (carvacrol), and Thy (thymol). * It indicates that the binary combination caused statistically (*p* < 0.05, Kruskal–Wallis test) a greater reduction in biofilm formation and showed an improved antibiofilm effect compared to both antimicrobials alone at the same concentrations. When the combination has an antibacterial effect similar to the antimicrobials alone, the difference in biofilm reduction is not significant. ±(standard deviation).

**Table 6 antibiotics-12-00756-t006:** Antibiofilm effect, which is shown in terms of metabolic activity reduction of oregano derivatives and biological silver nanoparticles (bioAgNP) compared to both antimicrobials individually against and KPC-producing *Klebsiella pneumoniae* (KPC) biofilms growth in microtiter plates, which were evaluated at preformed biofilm condition.

Bacteria	Antimicrobial Concentrations in Binary Combinations	Biofilm Reduction Caused by Combinations	Antibiofilm Effect of Combinations Compared to Antimicrobials Alone	Antimicrobial Concentrations Individually	Biofilm Reduction Caused by Antibacterials Alone
		Metabolic Activity Decrease (%)			Metabolic Activity Decrease (%)
**KPC**	**OEO + bioAgNP**		Improved	OEO at 0.30 mg/mL	39 ± 0.1
	0.30 mg/mL + 1.97 µg/mL	80 ± 1.1 *		OEO at 0.15 mg/mL	31 ± 1.1
	0.15 mg/mL + 0.98 µg/mL	46 ± 1.6 *		Car at 0.15 mg/mL	33 ± 0.8
	**Car + bioAgNP**		Similar	Car at 0.07 mg/mL	34 ± 0.6
	0.15 mg/mL + 1.97 µg/mL	66 ± 0.9		Thy at 0.06 mg/mL	15 ± 0.6
	0.07 mg/mL + 0.98 µg/mL	37 ± 1.1		Thy at 0.03 mg/mL	10 ± 1.4
	**Thy + bioAgNP**		Similar	bioAgNP at 1.97 µg/mL	55 ± 2.4
	0.06 mg/mL + 1.97 µg/mL	62 ± 3.2		bioAgNP at 0.98 µg/mL	0 ± 0.0
	0.03 mg/mL + 0.98 µg/mL	3 ± 0.9			
	**Car + Thy**		Improved		
	0.15 mg/mL + 0.06 mg/mL	74 ± 1.4 *			
	0.08 mg/mL + 0.03 mg/mL	53 ± 1.6 *			

Oregano derivatives: OEO (oregano essential oil), Car (carvacrol), and Thy (thymol). * It indicates that the binary combination caused statistically (*p* < 0.05, Kruskal–Wallis test) a greater reduction in biofilm formation and showed an improved antibiofilm effect compared to both antimicrobials alone at the same concentrations. When the combination has an antibacterial effect similar to the antimicrobials alone, the difference in biofilm reduction is not significant. ±(standard deviation).

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
