# Peer review of "Antibiofilm Effect of Biogenic Silver Nanoparticles Combined with Oregano Derivatives against Carbapenem-Resistant Klebsiella pneumoniae"

_antibiotics, 2023, doi:10.3390/antibiotics12040756_

Round 1

Reviewer 1 Report

Dear authors,

The manuscript entitled "Antibiofilm effect of biogenic silver nanoparticles combined with oregano derivatives against carbapenem-resistant Klebsiella pneumoniae” was to investigate the This research shows that OEO, Car, Thy, and bioAgNP, in combination, prevent biofilm formation and eradicate preformed biofilm by Enteroaggregative E. coli (EAEC 042) and carbapenemase (KPC)-producing K. pneumoniae.

It presents scientific relevance for the area of Plant Science and Medicine area.

After consulting www.sciencedirect.com and https://pubmed.ncbi.nlm.nih.gov/, publications were found for some authors involving the theme. However, you need to change some details/information in the Introduction, Material and Methods, Results, discussion and “conclusions”.

-line 136: the novelty of the research should be mentioned at the beginning. appropriate or in conclusions.

-the discussion in the introduction addresses both the food field and other fields where there are alternative antimicrobials with potential to combat resistant bacteria. I suggest a reorganization of the discussions with more force and distinctly on each field of interest.

-line 142: the explanation of the choice of some bacteria belongs to the methods chapter 4.1 (description of the strains, motivation of the choice, considerations for and against).

-line: improve the quality of the image marked C and D.

-in figure 1, I suggest debolding the letters. Please see all the figures.

-line 465, 796: please remove the space.

- line 808: ‟ant‟.

-line 811: please insert the refereces.

-in the figures, please review the font and size, they are disproportionate (eg. Fig. 3).

-please revise the spacing of the footnotes in the tables

Reviewer 2 Report

1-     Keywords is important part of article form but, in this article, they are too many, reduce them.

2-    Conclusion vs Conclusions.

3-    Conclusion section is long with many details (For example it should be without figures and without reference citation). Paraphrase it.

Reviewer 3 Report

Dear Authors

Thanks for submitting manuscript with title:  Antibiofilm effect of biogenic silver nanoparticles combined with oregano derivatives against carbapenem-resistant Klebsiella pneumoniae. The review of the aforementioned manuscript has been finished and there are some points about it which you could find at the below: 

1- The number of references, is too much (145) and its not common for original article and i think you have to reduce and delete some extra explanation in the text (for example in introduction section, the number of references is 74 and is too much).

2- The number of figures and tables are too much and you have too delete some of them.  

3- About the materials and methods section is need some revisions to get understanding, is very complicated to follow 

4- At the results section, again is very complicated and there are a lot of data for following .

5- at the conclusion, is too much and doesnt need, you have to revise and address just the main results .

Best Regards

Round 2

Reviewer 3 Report

Dear Authors

Thanks for your revised manuscript. The review of the revised manuscript has been finished and the revised is good and acceptable. 

Best Regards